# A Gabor Filter-Based Protocol for Automated Image-Based Building Detection

Hafiz Suliman Munawar [1], Riya Aggarwal [2], Zakria Qadir [3,*], Sara Imran Khan [4], Abbas Z. Kouzani [5] and M. A. Parvez Mahmud [5]

1 School of Built Environment, University of New South Wales, Kensington, NSW 2052, Australia; h.munawar@unsw.edu.au
2 School of Mathematics and Physical Sciences, The University of Newcastle, Callaghan, NSW 2308, Australia; riya.aggarwal@uon.edu.au
3 School of Computing Engineering and Mathematics, Western Sydney University, Penrith, NSW 2751, Australia
4 Faculty of Chemical Engineering, University of New South Wales, Kensington, NSW 2052, Australia; saraimrankhan17@gmail.com
5 School of Engineering, Deakin University, Geelong, VIC 3216, Australia; abbas.kouzani@deakin.edu.au (A.Z.K.); m.a.mahmud@deakin.edu.au (M.A.P.M.)
* Correspondence: z.qadir@westernsydney.edu.au

**Abstract:** Detecting buildings from high-resolution satellite imagery is beneficial in mapping, environmental preparation, disaster management, military planning, urban planning and research purposes. Differentiating buildings from the images is possible however, it may be a time-consuming or complicated process. Therefore, the high-resolution imagery from satellites needs to be automated to detect the buildings. Additionally, buildings exhibit several different characteristics, and their appearance in these images is unplanned. Moreover, buildings in the metropolitan environment are typically crowded and complicated. Therefore, it is challenging to identify the building and hard to locate them. To resolve this situation, a novel probabilistic method has been suggested using local features and probabilistic approaches. A local feature extraction technique was implemented, which was used to calculate the probability density function. The locations in the image were represented as joint probability distributions and were used to estimate their probability distribution function (pdf). The density of building locations in the image was extracted. Kernel density distribution was also used to find the density flow for different metropolitan cities such as Sydney (Australia), Tokyo (Japan), and Mumbai (India), which is useful for distribution intensity and pattern of facility point f interest (POI). The purpose system can detect buildings/rooftops and to test our system, we choose some crops with panchromatic high-resolution satellite images from Australia and our results looks promising with high efficiency and minimal computational time for feature extraction. We were able to detect buildings with shadows and building without shadows in 0.4468 (seconds) and 0.5126 (seconds) respectively.

**Keywords:** building detection; aerial image dataset; image processing; local feature extraction

## 1. Introduction

Remote sensing imagery has been used for a long time for building detection for various applications such as urban planning, estimation of population, mapping out building and or marketing perspectives. The operational methods developed over the years for building detection are semi-automated requiring the need of skilled personnel. The key role of operator is to identify rooftops of the buildings, defining the walls and borders which is tedious and expensive work. Manually marking of buildings in the aerial or satellite image has certain limitations [1]. It is possible to render buildings visible from various angles as they may not have an exact articulation. Other factors such as

vegetation, cloud cover and other infrastructure can affect building detection. Moreover, the image brightness and contrast may not be enough to classify infrastructures accurately and involve wide areas with multiple buildings. Also, buildings will never adhere to standard sizes or geometries. Various semi-automated systems have been developed; however, limited number of automated systems have been reported in the literature. These studies have reported limitation in obtaining high quality results when detecting large size building using aerial imagery [2]. These conventional methods show poor performance as the objects in urban areas have complex spectral and spatial characteristics. The objects are detected based on spectral information of individual pixels while neglecting spatial information. Researchers have invested efforts in developing approaches for automatic detection of different objects such as vegetative cover, building infrastructures, vehicles, and facilities with the help of satellite imagery. However, these systems still lack in dealing with the complexities of urban infrastructures. Therefore, to promote the development of automated visual building detection, it is crucial to develop high-resolution satellite image classification algorithms.

Researchers have been investigating different techniques for improving and simplifying automated image-based building extraction methods. Segl and Kaufmann [3] used an iterative process to assess joint supervised shape categorization along with unsupervised image segmentation and permits searching individual articles on satellite images at a high resolution. Molinier et al. [4] considered training by a self-organising map for the identification of boundaries of structures in satellite images. Benediktsson et al. (2003) [5] use boundary details to classify streets and roads within an urban area. They fed two kinds of data to two separate classifiers. Then, developers combined the High—resolution images and GIS output to detect buildings on high-resolution imagery, however, studies still require a training module. Benediktsson et al. [6] applied statistical methods to derive structural details from satellite images to detect urban areas and buildings. Li et al. [7,8] looked at satellite images that showed suburban regions to detect street networks. It employs the use of vegetation indices, cluster analysis, decomposing binary images, and graph theory. To have a very promising method, there is a need to find better techniques for multispectral image detection. Akçay and Aksoy [8] also suggested a novel technique for image detection using unsupervised segmentation in high-resolution satellite imagery with the application of diverse knowledge. Idrissa et al. [9] extracted roads and buildings by filtering edges using Gaussian blurring and based on the vegetation index. On evaluating the edges of two samples images that were taken from the similar area, the researchers noticed sudden differences. This is a much harder problem to be solved compared to detecting. Detecting building positions, however, can help to extract building shapes from the picture. Although more details are visible with the improved resolution of satellite images, the building detection is still difficult. Main reasons are the denseness and the complexness of the scene. Approaches such as Canny filter have been used for extracting the building contours based on local first order operator with subsequent searching for local maxima. However, the obtained results could be unsatisfactory as the resolution of satellites images changes abruptly. To achieve the automation of the process is complex as it requires a certain thresholding operation for optimizing contour detection. Alternatively, the Gabor filter can be used which is widespread in frequencies and orientation and can achieve optimal joint resolution in time and frequency. Analytical results have indicated that Gabor filter responses are stable even when the parameters selected are sub-optimally. Ruhang et al. [10] assessed the PV roof resources of residential buildings in an urban district. the remote sensing method was applied for extracting the information about the roof resources. Three methods can be applied to solve this problem a pixel-based analysis based on statistical approach or object-based analysis using expert knowledge or signal processing view method. Application of the Gabor filter method gave a fast and accurate result for extraction of the information of the residential buildings.

Shen et al. [11] assessed the building extraction method based on remote sensing image via Gabor filter and multi-orientation $\pi$ local binary pattern (LBP) operator. This method

was aimed to achieve better visualisation and improved urban planning. Initially, the Gabor filter was used to extract multi-dimensional texture features from the images. Multi-orientation $\pi$ LBP operator at various orientation was used to obtain training samples. The location and shape of the building were achieved by conducting pixel-level discrimination. Result showed 94% accuracy of building extraction and improved land management.

Local descriptors are important for object recognition and Gabor filters are efficient in extracting local features. Gabor filters have been used for different computer vision tasks such as invariant object recognition [12] and to detect building and road structures from satellite images. Idrissia et al. [13], used Gabor filters along with NDVI (Normalized Difference Vegetation Index) in SPOT5 images for extracting structure features from the images. Changes were observed by comparing the edges of the two images from the same locations. The advantage of using the Gabor filter is the ability to distinguish the spatial locality and orientation selectivity. Similarly, Zhao et al. [14] used Gabor filter-based edge detection method for remote sensing images. Two features of the Gabor filter i.e., optimal central frequency and optimal spectrum scale were evaluated. With the use of phase randomization and HSSIM index, the optimal central frequency was evaluated. the decision about optimal spectrum scale was carried out when the PSD values were at maximum at the special frequency. Two of each QuickBird, WorldView, and IKONOS images were tested for method validation. For the six images tested the edges were extracted with optimal central frequencies of 131, 149, 181, 171, 121, and 129 cycles/image and the optimal scales of 43, 47, 54, 55, 51, and 52 cycles/image, respectively. This method was able to achieve better edge detection in comparison to other methods. Average completeness of 81.79% average correctness of 65.72% and average F-measure values of 73% was achieved. The F-measure for all tested images was higher than 70% indicating the suitability of the method for the selected images.

Similarly, Wu et al. [15] estimated land type change by extracting residential areas from raster maps. The existing algorithm had limitations of low positional accuracy of recognized boundary and inclusion of misidentified objects. This was overcome by using an automatic recognition method based on the Gabor filter for obtaining residential boundary from the samples of three different areas. The obtained results showed higher integrity and precision and outperformed previous techniques. The Gabor filter has the advantage of a high degree of extraction, high accuracy, complete and precise boundary. Studies have shown that the Gabor filter is suitable for extracting the contour of residential areas from scanned raster topographic map with a resolution higher than 200 dpi. However, the extraction depends on parameter setting, direct extraction of the residual area from a map is not possible with the needed boundary of the residential area to be closer.

Hence, in this study, a probabilistic method based on the Gabor filter was applied to achieve accurate and precise detection of the buildings. The study aimed to monitor the difference between two different areas of detection. Furthermore, the area detection process was compared with different techniques and computational time was estimated with and without shadow. A small town i.e., Newcastle, Australia and then big metropolitan cities like Sydney are selected for the image dataset. Since the selected method depends only on local features, global details are not required. Therefore, it seems to be a smart idea to divide the metropolitan/urban area into parts and to detect buildings in them separately. The location of buildings can be detected in these areas by using a novel probabilistic framework such as Gabor filter [16–18]. Gabor filter is a band-pass filter selective to both orientation and spatial frequency. It is suitable for detecting local structural patterns from images and has been widely applied to texture analysis and object recognition. To detect, local vector features were first extracted from the given image using the Gabor filter, taking these vectors as observations. In other words, the building positions in each picture are modelled as a typical random variable and estimate their probability density function using interpretations [19–24]. The probabilities and mode of the expected pdf led to the construction of locations in the provided image. No training is required in any of these

steps. The system was tested on various satellite image sets and the building detection output is presented in the result section.

## 2. Methodology

### 2.1. Data Collection

The first step of conducting the research was to collect literature consisting of recent research papers which proposed building detection methods using deep learning. A detailed literature review was carried out for this study and a VOSVIEWER analysis was conducted based on the most used keywords in this research area, as shown in Figure 1 First, a basic set of keywords were formed which were: "Deep learning", "Image processing", "Gabor filter", "building detection" and "satellite datasets". Most used keywords in the recent literature related to the basic set of keywords were then retrieved. As shown in Figure 1 these keywords include: " Building detection", "Gabor filter", "satellite image", "change detection", "remote sensing", "classification performance" and "image retrieval". These keywords were used along with the base set of keywords to completely exhaust the database and get a maximum number of articles. Popular and widely used search engines were opted to retrieve research articles for the current study such as Scopus, Google Scholar, Science Direct, Elsevier, Springer, ACM and MDPI. These articles were reviewed to search for research gaps and limitations of the recent approaches so that the proposed method can target these gaps.

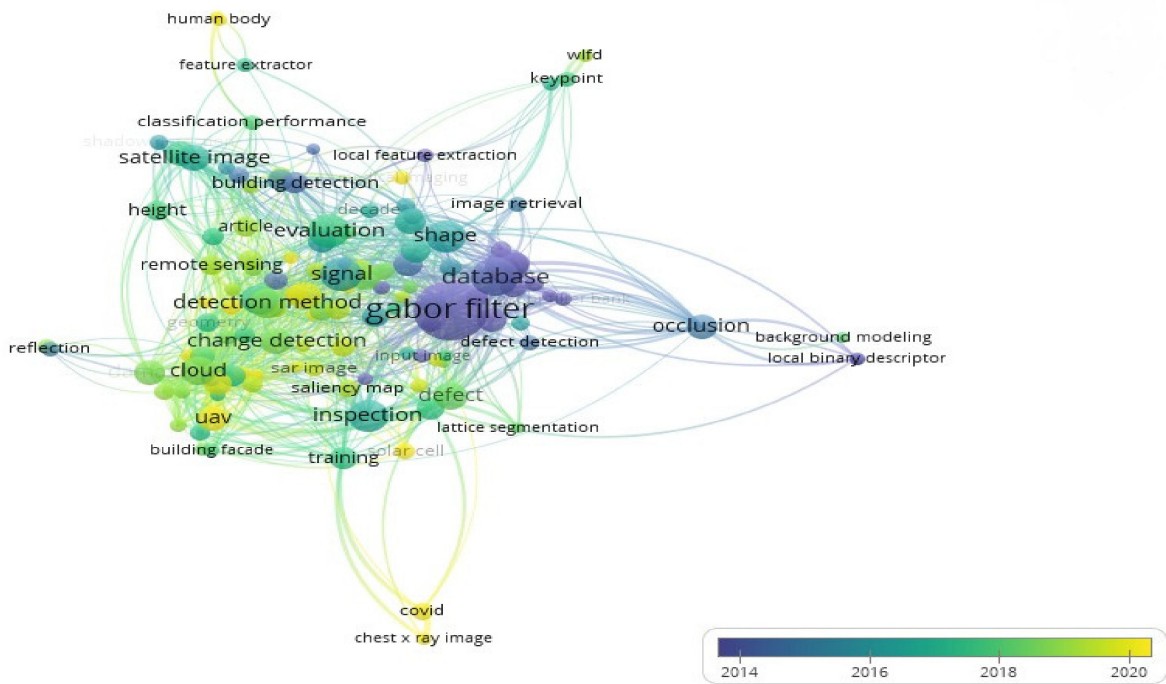

**Figure 1.** Keywords found in recent literature related to the base set of keywords.

### 2.2. Study Area and Training UAV Datasets

The study area selected was the Newcastle and Sydney regions of Australia. The UAV satellite-based data as overlayed on a map is shown in Figure 2.

In Figure 2, various architectures, building patterns, styles, illumination conditions and scene characteristics can be seen for the complete area of study. Figure 3 represents the complicated buildings present within the object area. These buildings can be grouped as per varying perspectives. For example, (i) medium and high rooftops (ii) tall and dense buildings (iii) small-sized rooftops (iv) green roofs of buildings (v) building roofs having playgrounds (vi) different building sides. The visual inspection method can easily detect the complex patterns of buildings, while the same is not possible through machine learning.

There are certain datasets presents online, namely the RSSCN7, PatternNet, UC Merced, AID, RSI-CB256, NWPU-RESISC45, Aerial Image Labeling and ISPRS labelling etc, which can present varying building patterns present around the globe [25–28]. These images are generally captured through satellite platforms, high spatial resolution imagery or aerial imagery. These object areas have a different training distribution that those of the open datasets. Hence, the first step in our study is to prepare a training setup for the dataset generated through satellite imagery, which is followed by generalizing the Seq-Net DL algorithm for the dataset collected from the area chosen for this study.

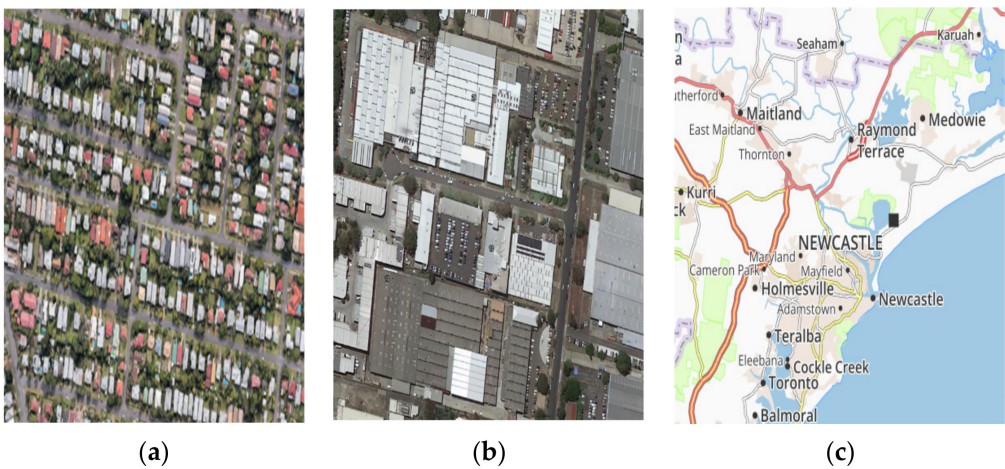

| (**a**) | (**b**) | (**c**) |

**Figure 2.** (**a**) Area 1 (**b**) Area 2 (**c**) Newcastle Map.

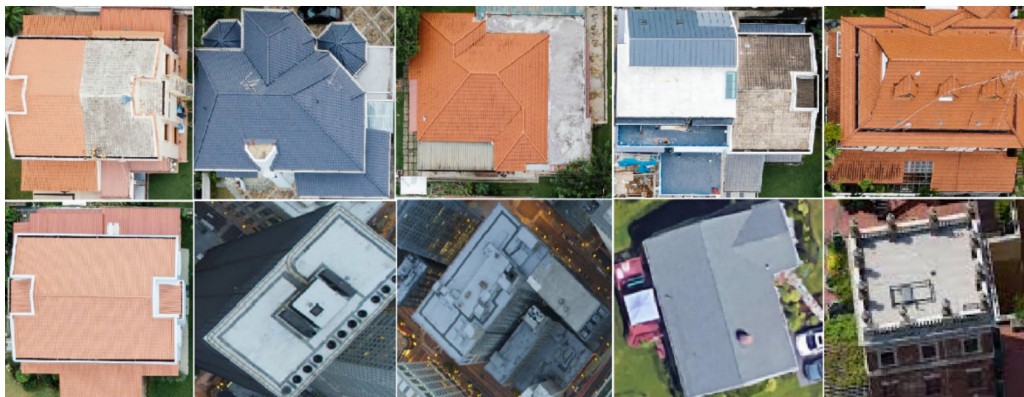

**Figure 3.** Aerial images of buildings with different perspectives in study area.

### 2.3. Classification of Datasets

To produce satellite datasets, firstly, all images were separated into two classes for training and were labelled as 'non-building' and 'building'. The biggest challenge in creating these datasets is the vast variety of patterns and details of the building and covered land areas. Figure 4 shows a sample dataset having 2200 images. Each image within the dataset consists of an original RGB image obtained from a satellite image and a labelled image containing two identified classes, both connected through a single ID. Whereas, Figure 5 shows the satellite image dataset example for metropolitan city (Sydney). In this study we have divided the annotated datasets into 2000 training sets, another set of 20% randomly selected sets for validation and another 220 test sets. Contrast to the training set, images from different regions are captured for the test set.

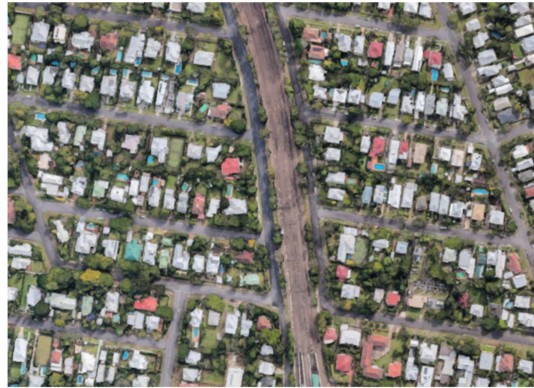
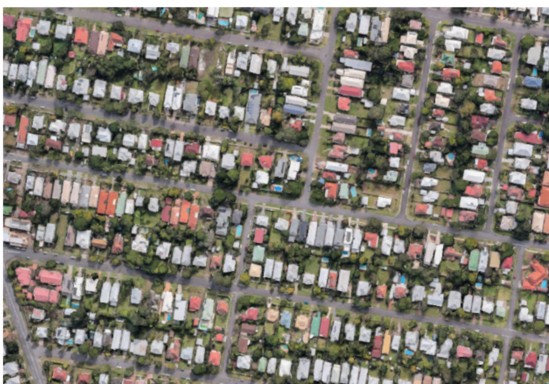

**Figure 4.** Satellite image dataset example for crowded small town (Newcastle).

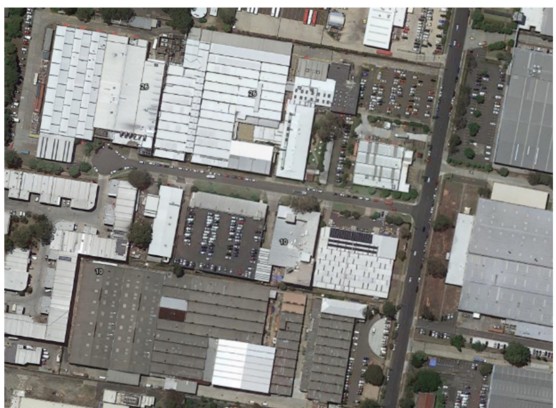
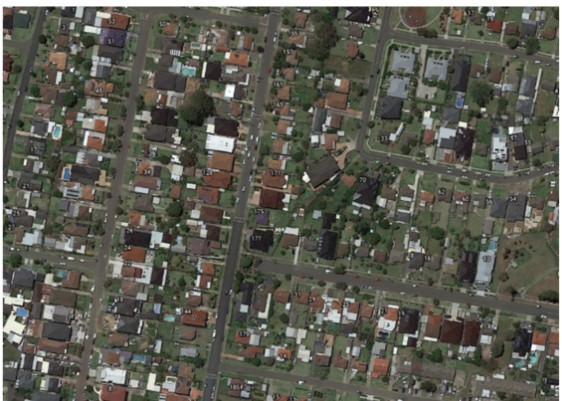

**Figure 5.** Satellite image dataset example for metropolitan city (Sydney).

## 3. Local Feature Extraction

The local features in an image were used to locate the areas. First, to extract the regions, the image was smoothed by median filtering [29]. The noise in the images is removed through this process. After this Gabor filtering was implemented in a separate direction. The local features is based on the maximum filter reactions. These measures in depth are explained below:

### 3.1. Gabor Feature Extraction

The Gabor features have been used commonly in image processing and analysis. The Gabor filter, originally developed by Dennis Gabor, is a linear filter mostly employed edge detection, surface evaluation, feature extraction, object recognition, and many other applications [30–32]. These filters possess optimum locality in the frequency and spatial domain effective for texture/surface mapping applications. These are types of bandpass filters that allow a certain frequency range while rejecting frequencies outside of it [33–35].

Mathematically, certain parameters influence how the Gaussian filter will operate and how it will respond to different feature elements. A 2-D Gabor filter can be regarded as a sinusoidal signal with a Gaussian wave modulating specific frequency and orientation [36–38]. The filter has two orthogonal components representing real and complex imaginary components. The two elements can be used individually or in a complex number. The equation is written as

$$F(x,y;\sigma,\vartheta) = \frac{1}{2\pi\sigma^2}\exp\left(-\frac{u^2+v^2}{2\sigma^2}\right)\exp\left(i2\pi\frac{u}{\lambda}\right)$$

where, $u = x \cos \vartheta + y \sin \vartheta$, and $v = -x \sin \vartheta + y \cos \vartheta$. $\lambda$ is the wavelength of the complex exponential signal, $\vartheta$ is the alignment of the normal to parallel lines of the Gabor filter, and $\sigma$ is the scale parameter or standard deviation of Gaussian envelope. These parameters control the size and shape of the Gabor function.

The Gabor filter can be used for testing the building edges in test samples. The response of the Gabor filter for the test image is given as below:

$$G(x, y; \sigma, \vartheta) = \frac{1}{2\pi\sigma^2} \exp\left(-\frac{u^2 + v^2}{2\sigma^2}\right) \cos 2\pi \frac{u}{\lambda}$$

where $G(x,y;\sigma,\vartheta)$ represents the maximum regions having similar characteristics with the filter. By using this information, the local feature points could be extracted. To do so, first, there is a need to search for the local maxima in $G(x, y; \sigma, \vartheta)$ for $x \in \{1, 2, \ldots, N\}$, $y \in \{1, 2, \ldots, M\}$.

### 3.2. Parameter Control Points

In the Gabor function the wavelength $\lambda$ manages the strips width. With the increase in the wavelength thicker strips are produced and with decrease in the wavelength the width of the strips decreases. The strips are thicker by retaining other factors and on increasing the wavelength from 70 to 100. The Gabor function is controlled by the theta $\vartheta$. When theta is zero degrees the position of the Gabor function is vertical. While the size of the Gabor filter is controlled by the sigma $\sigma$. The envelopes increase in width with inclusion of more stripes with larger bandwidth, conversely the width of the envelope reduces with small bandwidth [39–41]. The number of stripes can be enhanced by increasing the sigma to 25 and 45 in the Gabor function.

### 3.3. Gabor Local Feature Point Extraction

For extracting the local feature, the local maxima in $G(x, y; \sigma, \vartheta)$ for $x \in \{1, 2, \ldots, N\}$, $y \in \{1, 2, \ldots, M\}$ is assessed. If any pixel $(x_0, y_0)$ in $G(x, y; \sigma, \vartheta)$ has the largest value in its neighborhood,

$$(x_0, y_0; \sigma, \vartheta) > G(x_n, y_n; \sigma, \vartheta) \; \forall \; (x_n, y_n) \in \{(x_0 - 1, y_0 - 1), (x_0, y_0 - 1), \ldots, ((x_0 + 1, y_0 + 1))\},$$

Then it is called a local maximum. This location can be called a local distinctive feature. Later, the extent of the filter response $G(x_0, y_0; \sigma, \vartheta)$ is evaluated. The local maximum $(x_0, y_0)$ is denoted as representative local feature point if meets the condition $G(x_0, y_0; \sigma, \vartheta) > \alpha$. Using $\alpha$ by adopting Otsu's method on $G(x, y; \sigma, \vartheta)$ is obtained adaptively for handling different images separately. Consequently, in future estimates, the poor candidate local feature points are deleted. A weight is allocated to further represent each local feature point of the candidate. The first threshold is $G(x, y; \sigma, \vartheta)$ with $\alpha$ following a binary image $Im(x, y; \sigma, \vartheta)$. In the image if pixels correspond to one it relates to robust responses. Thus, the linked pixels to $(x_0, y_0)$ in $Im(x_0, y_0; \sigma, \vartheta)$ are obtained. The Figure 6. Shows the building area detection using Gabor filter results on a sample satellite images, in a binary image two pixels are connected to each other if the value of the pixel is one or are connected by a path [11].

All the pixels are connected to $(x_0, y_0)$, are obtained by their sum as the weight $w_0$ are assigned. Thus, the selected local feature point has more weight if more pixels are attached. It is expected that the local features of the candidate to reflect area features like building groups. Sadly, not all the local feature points of the candidate will provide accurate area statistics. Therefore, the selected local feature points having weight $w_0 < 25$ pixels are discarded. Although the weight threshold value applied is similar for all images with different characteristics, it is essential to adapt this to the test image. Lastly, for the given direction $\vartheta$, the local characteristic points were obtained. Figure 7 details the working of the Gabor filter method.

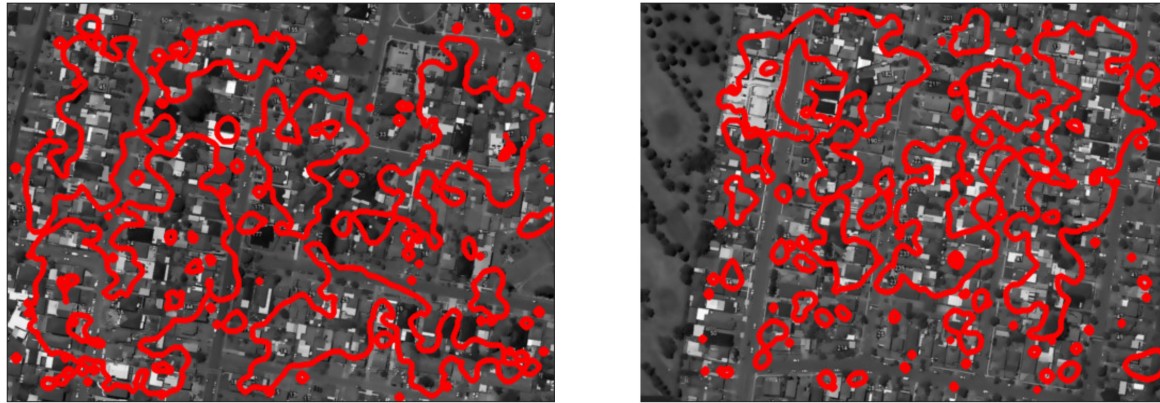

**Figure 6.** Building area detection using Gabor filter results on a sample satellite images.

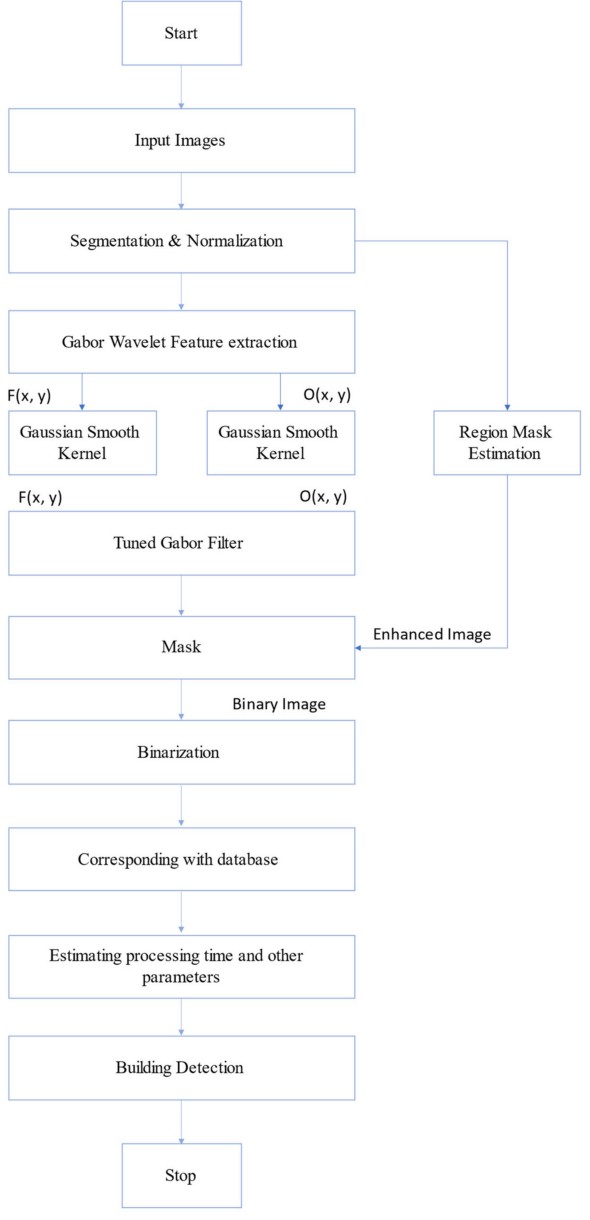

**Figure 7.** Gabor Filter's Building Detection Process.

## 4. Building Detection

After the experimental validation of the obtained results, the satellite images of the selected region will be discussed in detail in this section. The Figure 8a,b depicts the results of building being detected in the Newcastle and Sydney region by the local feature extractions method. As evident from the images all the building are detected with precision with this method [42–44]. The two-test image of size $512 \times 512$ are selected, our area detection method labelled different regions. The probabilistic building detection along with decision fusion was applied on the selected regions and results were depicted with grayscale images in Figure 8. From the obtained results it was evident that the building detection was carried out accurately with precision. In the next step, the current method was applied on the 32 images for testing. Among all these images, 24 images were taken from the Newcastle region, five from Sydney (CBD), and four were taken over Sydney residential away from CBD. A wide range of geographical conditions were covered in the images with a range of 250 to 300 buildings detected.

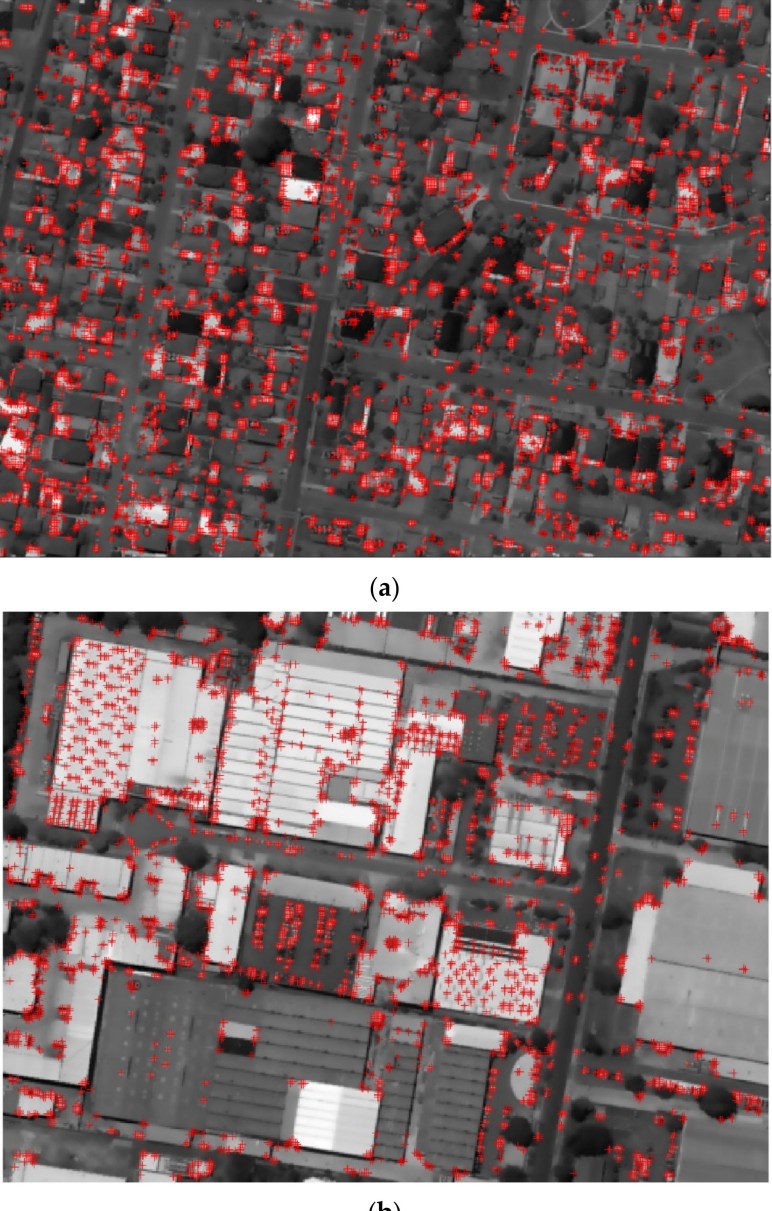

(**a**)

(**b**)

**Figure 8.** (**a**) Gray scale image with local feature extraction on a sample satellite images. (**b**) Gray scale image with local feature extraction on a different sample satellite images.

The above procedure was applied directional to obtain a sum of *l* local feature positions as $(x_l, y_l)$ with their weights $w_l$ for $l \in \{1, \dots, L\}$. The local characteristics to be located at the edges of the building was expected. The Figure 8 depicts the sample of testing satellite image, with the presence of functional point on the image edges. Other researchers have indicated more sophisticated methods for extracting feature points [31,32]. However, to detect urban areas, there is no need to extract the local point. As their collection is used the efficiency of the area detection process is affected by the lack of a few local characteristics. Final building detection images are shown in Figure 9.

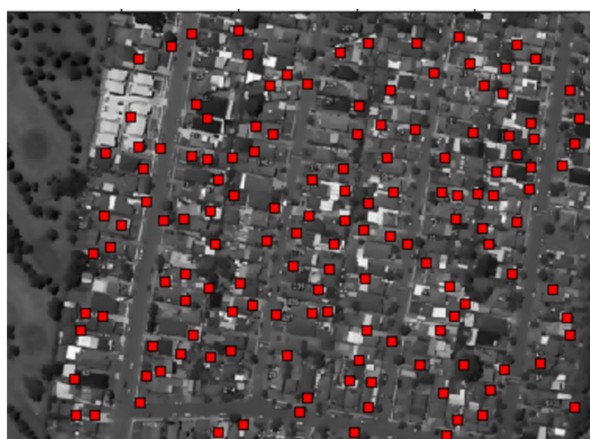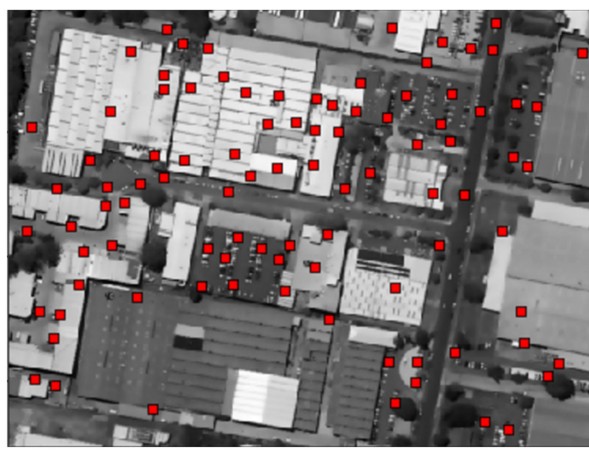

**Figure 9.** Buildings detected by the proposed method.

## 5. Density Flow

Kernel density distribution is used to find the density flow for different metropolitian cities such as Sydney (Australia), Tokyo (Japan), and Mumbai (India). Since we have *L* features, there will be *L* possible building centers with coordinates $(x_l, y_l)$. We form the kernel density matrix by using following formula

$$K(x,y) = \sum_{l=1}^{L} \frac{1}{2\pi\sigma^2} \exp\left( -\frac{(x-x_l)^2 + (x-x_l)^2}{2\sigma^2} \right)$$

Here $\sigma$ is the parameter for kernel proximity for each local feature. This means, we give the maximum value/vote to the possible building center in $(x_l, y_l)$ coordinates. We also give vote to its neighboring locations in decreasing order (using a Gaussian function).

After finding the voting matrix, we locate buildings by detecting local maximums of the voting matrix, $K(x,y)$. Possible building locations are the points where votes make a maximum (where probability is maximum) in $K(x,y)$. We detect the $(x_l, y_l)$ location from $K(x,y)$ as a possible building.

The biggest challenge in today's world is overpopulation, however, the impact of the overpopulation are the issues that many countries like Japan and India are facing. Nevertheless, population density is not something that is easily controllable, which leads to the issues that these densely populated cities face such as water supply and housing shortages. This is it important to analyse the urban density of the cities [26,27,45]. With approximately 28,000 people per square kilometer (73,000 per square mile), Mumbai is one of the most densely populated areas in the world. Sydney is having 1171 persons per square kilometre. In comparison, Tokyo counts 6158 people per square km. On the other hand, Mumbai is known for its colonial-era buildings, soviet-style offices and two UNESCO world heritage sites. Mumbai, Tokyo and Sydney are one of the most important cities in the world currently. Few density map results are shown for 3 cities in Figures 10–14. Final evaluation summary of each city is shown in the Table 1.

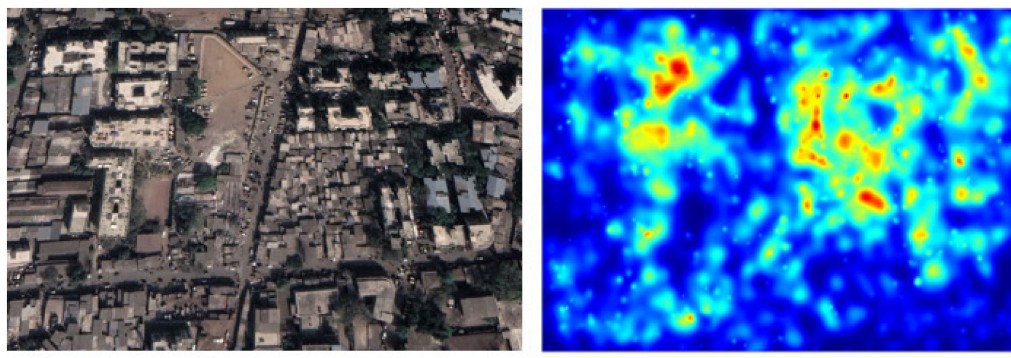

**Figure 10.** Density Map for Mumbai city (India).

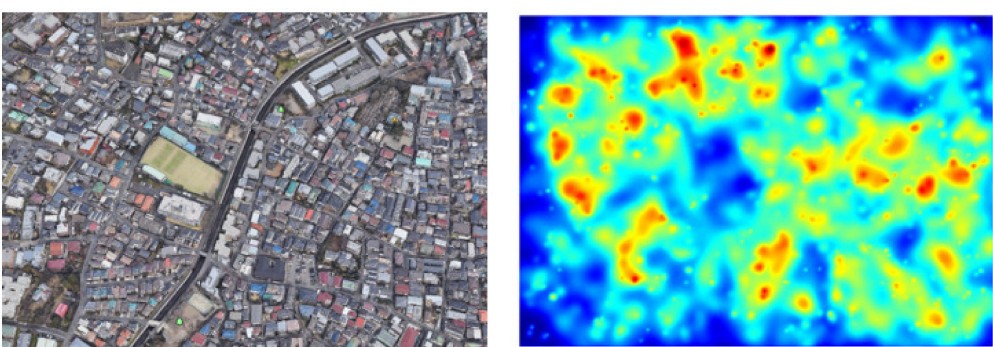

**Figure 11.** Density Map for Tokyo city (Japan).

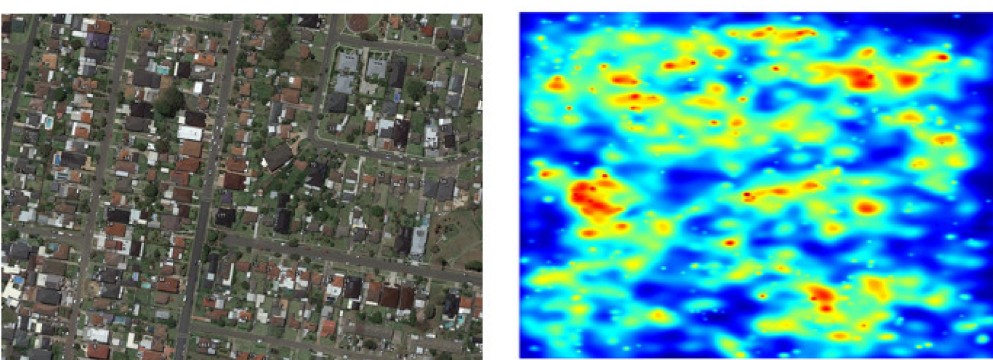

**Figure 12.** Density Map for Sydney city (Australia).

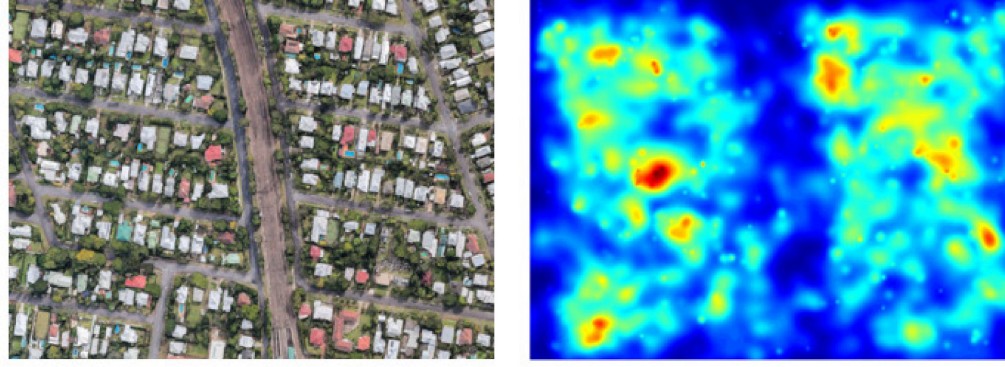

**Figure 13.** Density Map for Newcastle (Australia).

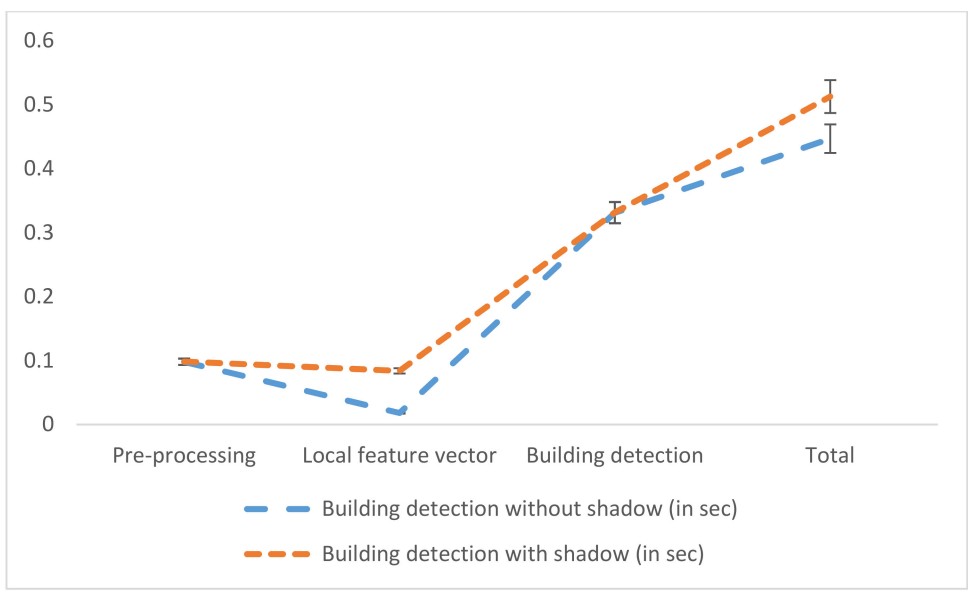

**Figure 14.** Building Detection computational time comparison.

**Table 1.** Density recognition percentage.

| Sydney | Tokyo | Mumbai | Newcastle |
|---|---|---|---|
| 0.90 | 0.88 | 0.82 | 0.93 |

## 6. Computational Time

Computational time is another important factor that needs to be considered for this technique. The computational time of the selected technique can be assessed by considering the CPU timing in the satellite image with a pixel of 512 × 512. A laptop with Intel Core i7-7600U CPU, 2.80-GHz processor and 16 GB RAM was used for recording the computational time. All the data was coded using the MATLAB platform. The time required for crowded area detection was determined in two conditions, one is with shadow prediction and the second is without shadow. More information about computation timings can be found in Table 2.

**Table 2.** Computational Time comparison for building detection without shadow and with shadow.

| Steps | Building Detection without Shadow (in Sec) | Building Detection with Shadow (in Sec) |
|---|---|---|
| Pre-processing | 0.0979 | 0.0982 |
| Local feature vector | 0.0179 | 0.0837 |
| Building detection | 0.3310 | 0.3315 |
| Total | 0.4468 | 0.5126 |
| Accuracy (Building detection) | 91.2% | 93.5% |

Table 3 Computational Time comparison for building detection using different algorithm.

Figure 15 depict the computational time (with error bars indicating standard deviation) comparison between the different methods with Gabor filter. It was found that the lowest computational time was observed for the Gabor filter with and without shadow. Followed by the computational time of Previt, Sobel and Robert. Maximum computation time was observed for Canny and Zerocorss. Hence, building detection based on Gabor Filter is time efficient.

**Table 3.** Computational Time comparison for building detection.

| Methods | Building Detection without Shadow (in Sec) | Building Detection with Shadow (in Sec) |
|---|---|---|
| Canny | 0.979 | 0.982 |
| Robert | 0.779 | 0.837 |
| Previt | 0.510 | 0.575 |
| Sobel | 0.668 | 0.712 |
| Zerocorss | 0.889 | 0.979 |
| Gabor Filter | 0.4468 | 0.5126 |

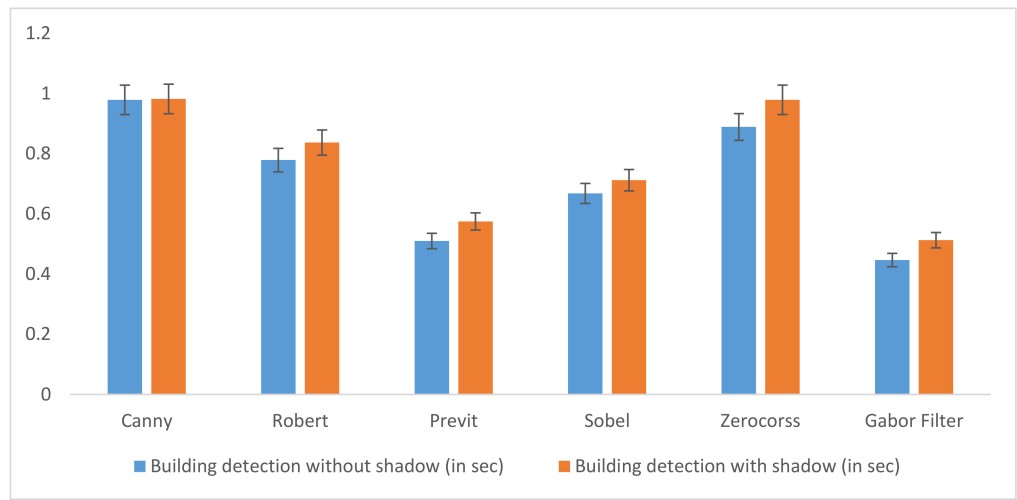

**Figure 15.** Building Detection's Computational Time Comparison.

## 7. Conclusions

Hence, this study focused on two different areas of detections with very high-resolution satellite imagery (small cities and metropolitan cities) and building detection based on Gabor features. This would be the key step in the monitoring of the difference between the two areas of detection. The current method relied on Gabor filtering for local feature point extraction. Informing a spatial voting matrix, local feature points were used. The city area in each satellite image was detected using an optimal decision-making method.

Promising results were obtained based on our system after conducting extensive research. The area detection process was rapid and reliable in comparison with the existing algorithm. The computational time required for crowded area detection was faster without shadow as compared to its prediction with shadow. The performance may also be improved by applying probabilistic relaxation. The automatic method of damage detection used in this study is a reliable technique which can be employed during disaster event and facilitate in humanitarian aid. However, additional calculations and analysis are required. Hence the proposed method is a robust tool for the automated building detection system. In future, further work will be carried out to detect the type of area by analyzing the vote matrix characteristics (e.g., dense, homogenous, and well structured). In future we can use GAN-based methods for image-to-image translation and to explore pixel wise details using Deep Learning.

**Author Contributions:** Conceptualization, H.S.M., R.A., Z.Q. and S.I.K.; methodology H.S.M., Z.Q. and S.I.K. and software, H.S.M., R.A.; validation, S.I.K.; formal analysis, H.S.M., Z.Q. investigation, H.S.M., Z.Q. and S.I.K. resources, A.Z.K.; data curation, M.A.P.M.; writing—original draft preparation, H.S.M., Z.Q. and S.I.K. funding acquisition, M.A.P.M. All authors have read and agreed to the published version of the manuscript.

**Funding:** This research received no external funding.

**Institutional Review Board Statement:** Not applicable.

**Informed Consent Statement:** Not applicable.

**Data Availability Statement:** Not applicable.

**Conflicts of Interest:** The authors declare no conflict of interest.

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
