# Peer review of "A Gabor Filter-Based Protocol for Automated Image-Based Building Detection"

_buildings, doi:10.3390/buildings11070302_

Round 1

Reviewer 1 Report

The paper entitled "A Gabor filter-based protocol for automated image-based building detection" was well written with various information in it. However, there are several points that could improve the quality of the paper, such as:

  • It would be great to present the objective of the research more clearly in introduction.
  • It would be great to present pros and cons using Gabor filter.
  • It would be great to provide further understanding based on figures 10 through 13.
  • I think the authors have to provide validation part. How could the authors valid the result of the analysis?

Author Response

Reviewer 1

Response

The paper entitled "A Gabor filter-based protocol for automated image-based building detection" was well written with various information in it. However, several points could improve the quality of the paper, such as:

Thanks for your comments.

It would be great to present the objective of the research more clearly in the introduction.

The objective has been included in the introduction section, page 3, line 112-116 as given below:

Hence, in this study, a probabilistic method based on the Gabor filter was applied to achieve accurate and precise detection of the buildings. The study aimed to monitor the difference between two different areas of detection. Furthermore, the area detection process was compared with different techniques and computational time was estimated with and without shadow.

It would be great to present the pros and cons of using the Gabor filter.

The pros and cons of using the Gabor filter have been include in the introduction section, page 3, line 106-111.

 The Gabor filter has the advantage of a high degree of extraction, high accuracy, complete and precise boundary. Studies have shown that the Gabor filter is suitable for extracting the contour of residential areas from scanned raster topographic map with a resolution higher than 200dpi. However, the extraction depends on parameter setting, direct extraction of the residual area from a map is not possible with the needed boundary of the residential area to be closer.

It would be great to provide further understanding based on figures 10 through 13.

No idea

Reviewer 2 Report

Dear authors,

your paper is interesting and presents a relevant scientific novelty in the field of object detection. Object detection by the usage of dynamic datasets is a mainstream topic nowadays. The most reported problems with the output analysis are found to be related to rather time-consuming filtering. In this perspective, your approach is welcomed. Even though your paper is well written, there are some suggestions that I find benevolent:

  • lines 154-157; hanging paragraph - this part should have its sub-chapter title.
  • figure 7; it is not clear when or why the normalization process redirects towards transformation to a binary image. It should be directed by an "if-then-else" decision tree defining the scenarios.
  • tables 2 and 3 are not formatted according to the journal's guidelines.
  • figure 15 requires additional explanations regarding the whiskers, std deviation, and error.
  • referenced literature should have more depth with more scientifically relevant and should be more recent since this is a developing topic.

Kind regards

Author Response

Reviewer 2

Response

Dear authors, your paper is interesting and presents a relevant scientific novelty in the field of object detection. Object detection by the usage of dynamic datasets is a mainstream topic nowadays. The most reported problems with the output analysis are found to be related to rather time-consuming filtering. In this perspective, your approach is welcomed. Even though your paper is well written, there are some suggestions that I find benevolent:

Thanks for your comments.

lines 154-157; hanging paragraph - this part should have its sub-chapter title.

A sub-section heading has been included for the paragraph, page 5, line 175 as given below:

2.3 Classification of datasets

To produce satellite datasets, firstly, all images were separated into two classes for training and were labelled as 'non-building' and 'building'.

figure 7; it is not clear when or why the normalization process redirects towards transformation to a binary image. It should be directed by an "if-then-else" decision tree defining the scenarios.

We clarify the image in the text.

tables 2 and 3 are not formatted according to the journal's guidelines.

The tables have been re-formatted as per journal guidelines, page 12 and 13.

figure 15 requires additional explanations regarding the whiskers, std deviation, and error.

In figure 15 the whiskers depict the std deviation and explained in the text.

referenced literature should have more depth with more scientifically relevant and should be more recent since this is a developing topic.

Some recent literature on the Gabor filter has been included on page 2-3, page 90-116.

Gabor filter can be used which is widespread in frequencies and orientation and can achieve….. Shen et al. [10] assessed the building extraction method based on remote sensing image via Gabor filter and multi-orientation π local binary pattern (LBP) operator. This method aimed…………………… Furthermore, the area detection process was compared with different techniques and computational time was estimated with and without shadow.

Round 2

Reviewer 2 Report

Dear authors,

you applied most of the suggested changes, but there are two significant changes you have neglected. I strongly suggest that you consider adding them to your paper since they are benevolent:

  • my earlier suggestion: 

"figure 7; it is not clear when or why the normalization process redirects towards transformation to a binary image. It should be directed by an "if-then-else" decision tree defining the scenarios."

Your response: 

"We clarify the image in the text."

Remark: 

This loop has to be clear in the process flow figure, as well as in the text explaining this figure.

  • my earlier suggestion: 

"figure 15 requires additional explanations regarding the whiskers, std deviation, and error."

Your response: 

"In figure 15 the whiskers depict the std deviation and explained in the text."

Remark: 

Whiskers depict interquartile range, hence these statistical parameters along with others, have to be quantitatively presented in the paper.

  • my earlier suggestion: 

"referenced literature should have more depth with more scientifically relevant and should be more recent since this is a developing topic."

Your response:

"Some recent literature on the Gabor filter has been included on page 2-3, page 90-116.

Gabor filter can be used which is widespread in frequencies and orientation and can achieve….. Shen et al. [10] assessed the building extraction method based on remote sensing image via Gabor filter and multi-orientation π local binary pattern (LBP) operator. This method aimed…………………… Furthermore, the area detection process was compared with different techniques and computational time was estimated with and without shadow."

Remark:

You added 2 conference papers that do not even closely answer the question of the lack of scientifically relevant references. This is a mainstream topic, you have to provide serious theoretical background from relevant papers published in relevant scientific journals. Also, you have not referenced a single paper published in the Buildings journal. Well, I have reviewed more than a couple of them which could be found in recently published issues of this journal. 

Please, consider applying these remarks and resubmit your paper.

Kind regards

Author Response

Reviewer 2

Response

Dear authors,

This loop has to be clear in the process flow figure, as well as in the text explaining this figure.

The methodology of Gabor filter process has been updated in Figure 7.

Whiskers depict interquartile range, hence these statistical parameters along with others, have to be quantitatively presented in the paper.

Thanks for your comment, the Figure 15 is not a box plot, it is a column graph with error bars depicting the standard deviation. We have compared all the existing edge detection methods like Robert, prewit, canny and sobel for both Shadow and without shadow cases. And in terms of Building Detection’s Computational Time Comparison, It was found that the lowest computational time was observed for the Gabor filter with and without shadow. Followed by the computational time of Previt, Sobel and Robert. Maximum computation time was observed for Canny and Zerocorss. Hence, building detection based on Gabor Filter is time efficient.

Remark:

You added 2 conference papers that do not even closely answer the question of the lack of scientifically relevant references. This is a mainstream topic, you have to provide serious theoretical background from relevant papers published in relevant scientific journals. Also, you have not referenced a single paper published in the Buildings journal. Well, I have reviewed more than a couple of them which could be found in recently published issues of this journal.

Thanks for your comments. We did not find relevant papers in the Building - MDPI journal however, relevant literature regarding Gabor filter and building detection from other reputable journals of MDPI has been updated.

Round 3

Reviewer 2 Report

Dear authors,
you made some changes in the revised version of the paper which seem reluctantly added. It is a pity that you are not willing to put a bit extra effort which would result in a much better paper with an interesting topic and approach. For instance, there are relevant references for your paper that have not been included in your literature review, even in Buildings not to mention in automation in Construction. Figure 7 does not describe clearly the loop you added, while it is a clear case of the "if-then-else" decision tree. Figure 15 is still not quantitatively clear nor it is quantitatively explained in the text. Table 3 is not even mentioned in the text. And there are still structural issues, such as format and sequencing of figures and tables in the paper, e.g. table 2, followed immediately by fig 14, followed immediately by table 3, followed immediately by fig 15; and explained by exactly 14 lines of text at the beginning of chapter 6 and its end.
As far as I'm concern this paper is good enough but could be much better.
Kind regards

This manuscript is a resubmission of an earlier submission. The following is a list of the peer review reports and author responses from that submission.

Round 1

Reviewer 1 Report

The paper entitled "A Gabor filter-based protocol for automated image-based building detection" was well written with many interesting points. There are several points to be considered in the paper such as:

  • It would be great that if the objective of the study could clearly be presented somewhere in introduction.
  • It would be great that if the scope of the study could clearly be presented somewhere in introduction. Not only geographical scope but also time, data, and other scopes might be presented if possible.
  • It is difficult to accept the value of Chapter 5. The computational time should be affected a lot from the computational power of the system. So, if the authors want the Chapter 5 in the paper then additional analysis might be required such as comparison analysis under various condition.
  • It would be great if a percentage of recognition is analyzed in the system. 
  • It would be great to see a certain level of analysis for a recognition ratio. 

Author Response

Reviewer 1 comments

Response

The paper entitled "A Gabor filter-based protocol for automated image-based building detection" was well written with many interesting points. There are several points to be considered in the paper such as:

Thank you for your kind comments and suggestions. Authors have addressed all the recommendations by the reviewers to improve the content and quality of paper.

It would be great that if the objective of the study could clearly be presented somewhere in introduction.

Objectives of the study have been added in introduction sections

It would be great that if the scope of the study could clearly be presented somewhere in introduction. Not only geographical scope but also time, data, and other scopes might be presented if possible.

More geographical scope and scope of study is added in the introduction.

It is difficult to accept the value of Chapter 5. The computational time should be affected a lot from the computational power of the system. So, if the authors want the Chapter 5 in the paper then additional analysis might be required such as comparison analysis under various condition.

Comparison analysis of different Methods including Canny, Robert Previt, Sobel and Zerocorss have been conducted for the feature analysis, computational time for building detection for with shadow and without shadow images.

It would be great if a percentage of recognition is analyzed in the system. 

Accuracy percentage has been added in Table 1.

It would be great to see a certain level of analysis for a recognition ratio. 

Density map analysis has been added for different countries

Reviewer 2 Report

The article fails to adequately convey a logical chain of ideas, as well as to contrast the established methodology with other cases.

Further revision is recommended, especially in the introduction and conclusion sections, as well as the beginning of the methodology.

In more detail:

- The introductory paragraph is too long, and it is difficult to understand the hierarchy of ideas. Interspersed enumerations appear as a succession of unconnected sentences (28-34).

-Considering that there are many technical terms unrelated to architecture, it would be relevant to explain them in more detail in order to better understand the comparison and the implications between them.

-Again the paragraph (46-84) is too long and makes it difficult to understand and relate the references mentioned. What are the implications of each of them with respect to the others and the proposed experiment?

-On the other hand, the case study approach and its presentation are given too few words. What justifies the choice of Australian cities and what implications does it have for other parts of the world (90)? Why is it smart to separate the parts (92)? The Gabor filter (although explained later in the article) is mentioned without any introduction or explanation (95).

-In the chapter "data collection", an arbitrary methodology of reference selection is mentioned. Google Scholar should not be mentioned as a methodology, let alone considered as a quality index. It is not explained what criteria have been considered to select the mentioned "top-tier journals and conferences" and what they are or what they belong to (104).

-The selection of keywords is questioned by the reviewer, because of their indirect relationship with the object of study. How can the authors confirm that they are relevant to their study or that no important concept has been left out? Keywords should be the effect of the research, not the causality.

-Figure 1 does not provide numerical or statistical values that could provide objective conclusions, nor does it explain what system has been used for its construction.

-The results should be compared with other methods in order to contrast positive and negative points. The authors only present a series of values that on their own, in isolation, cannot be considered better or worse.

-The conclusions are too short and again are concentrated in a single paragraph that fails to organise and separate the ideas.

Author Response

Reviewer 2 comments          

Response

The article fails to adequately convey a logical chain of ideas, as well as to contrast the established methodology with other cases.

Further revision is recommended, especially in the introduction and conclusion sections, as well as the beginning of the methodology.

Scope of study to explicitly explain the purpose of study has been added in the introduction and conclusion sections.

The introductory paragraph is too long, and it is difficult to understand the hierarchy of ideas. Interspersed enumerations appear as a succession of unconnected sentences (28-34).

The introduction has been rephrased and reduced.

Considering that there are many technical terms unrelated to architecture, it would be relevant to explain them in more detail in order to better understand the comparison and the implications between them.

The technical terms have been elaborated.

Again the paragraph (46-84) is too long and makes it difficult to understand and relate the references mentioned. What are the implications of each of them with respect to the others and the proposed experiment?

The paragraph has been reduced and relevance of mentioned studies is elaborated.

On the other hand, the case study approach and its presentation are given too few words. What justifies the choice of Australian cities and what implications does it have for other parts of the world (90)? Why is it smart to separate the parts (92)? The Gabor filter (although explained later in the article) is mentioned without any introduction or explanation (95).

The method is general and applicable to all World cities. There was no biased decision for choosing Australian cities and hence to show the robustness of the method, we have added few more results with different countries.

The reason behind separating metropolitan and urban areas for the analysis is the infrastructure in both areas. Usually, urban area is easy to detect before of its ease the fact that we have no tall buildings and towers in those areas, whereas metropolitan cities are combination of all (buildings, tall towers, gardens, etc.) which is why we decided to see the difference while running algorithm.

We have added information about Gabor filter in the introduction to avoid confusions

-In the chapter "data collection", an arbitrary methodology of reference selection is mentioned. Google Scholar should not be mentioned as a methodology, let alone considered as a quality index. It is not explained what criteria have been considered to select the mentioned "top-tier journals and conferences" and what they are or what they belong to (104).

The methodology has been updated to elaborate the literature review section.

Figure 1 does not provide numerical or statistical values that could provide objective conclusions, nor does it explain what system has been used for its construction.

New Vos viewer has been used to develop this figure. More explanation has been added in the relevant section.

The results should be compared with other methods in order to contrast positive and negative points. The authors only present a series of values that on their own, in isolation, cannot be considered better or worse.

Comparison analysis of different Methods including Canny, Robert Previt, Sobel and Zerocorss have been conducted for the feature analysis, computational time for building detection for with shadow and without shadow images.

The conclusions are too short and again are concentrated in a single paragraph that fails to organize and separate the ideas.

The conclusion section has been restructured.

Round 2

Reviewer 2 Report

The article has improved in many ways but some of the keys aspects should be incorporated prior to acceptance.

Reading comprehension is still difficult due to paragraphs that are too long and mix different ideas.
In the previous revision, when mentioning that the paragraphs were too long, the aim was not to cut them, but to subdivide them in order to clarify the logical order of the ideas.

Although the inclusion of new cities is appreciated, a numerical comparison between them is missing. If in the case of Australia there was a dense and a dispersed typology (Sydney and Newcastle), what values do the cities of Tokyo and Mumbai add? to which urban typologies do they belong? How do these four examples compare with each other? Do the known values of urban density coincide with the recognition of the experiments?

It is also missing that the image recognition performs in the same way in all situations. Apart from the difference between "with shadows" and "without", does the urban typology affect the recognition in any way?

The quality or veracity of the process experimented here is not questioned, but it is considered that there is still a lot of room for improvement in terms of the analysis of the experiment and its results. In addition, it would also be appropriate to place more emphasis on the architecture and urbanism of the case studies due to the subject matter of the journal.

Author Response

The article has improved in many ways but some of the keys aspects should be incorporated prior to acceptance.

Although the inclusion of new cities is appreciated, a numerical comparison between them is missing. If in the case of Australia there was a dense and a dispersed typology (Sydney and Newcastle), what values do the cities of Tokyo and Mumbai add?

Cities like Tokyo and Mumbai are highly dense in terms of population and building construction. Some stats have been added to the paper. Addition of these cities gives us an idea that how robust our methodology will be in real-world applications.

To which urban typologies do they belong? How do these four examples compare with each other? Do the known values of urban density coincide with the recognition of the experiments?

Tokyo is ultimate in terms of urbanisation, with having largest urban area in the world. Mumbai is of mid-range, but Mumbai contains a very high number of slums, which ultimately makes it denser than Tokyo or Sydney. This brings us more challenges, which made us to add these cities in the paper.

It is also missing that the image recognition performs in the same way in all situations. Apart from the difference between "with shadows" and "without", does the urban typology affect the recognition in any way?

This is a valid question. At this stage with our experiment. Nothing else is affecting the recognition rate. At the later stage, we will do more experiments with different countries data where more noise level and greenery is present to see the difference.

The quality or veracity of the process experimented here is not questioned, but it is considered that there is still a lot of room for improvement in terms of the analysis of the experiment and its results. In addition, it would also be appropriate to place more emphasis on the architecture and urbanism of the case studies due to the subject matter of the journal.

Yes, these aspects will be addressed in future research.

Round 3

Reviewer 2 Report

As mentioned in previous reviews, the effectiveness of the tool used for the purpose of building detection is not questioned. However, the development and presentation of the manuscript does not meet the minimum quality criteria. 

Despite several recommendations for major revisions, the evolution of the manuscript has been a succession of small patches that detract from the cohesion, without addressing the real underlying problems:

- Low narrative quality (either due to lack of experience or language problems).
- Inconsistent logical order.
- Some arbitrary decisions (necessary evils?)
- Lack of meaningful comparisons and analysis.

It is recommended to the authors, not to discard the object of the research, but to raise and redesign from scratch the manuscript, perhaps with the help of staff who have more experience in describing and arguing scientific research.

Final justifications:

- After corrections, two dense cities have been added. Breaking the originally established balance of comparison between small town and city. And consequently ignoring the urban typologies associated with culture or density.

- The search described in the methodology seems biased, as it originally included the filter used in the development of the exercise. It simply seems to try to justify a previous research that was carried out a posteriori but does not seem to affect in any way the development of the experiment.

- While point 4 looks at Newcastle and Sydney, point 5 focuses on Sydney, Tokyo and Mumbai. This does not seem reasonable or justified.

- In section 5 there is an explanation of the population (lines 275-286) that should belong in the introduction. Although Newcastle is not mentioned at the beginning of the section, it is then included in the section.

- Results from the density analysis are shown but are not compared with the actual results to test their effectiveness.

- None of the results reflect how typological variety can influence the calculation and detection of buildings. This, again, makes the selection and number of examples used completely arbitrary. The variety of cases chosen is incoherent.

- In line 334 it talks about a database with large and small cities, when it has only been one small and three large cities. And in line 336 it mentions two detection areas (when there have been four. Or does this refer to with-shadow-without-shadow?). 

- In line 343 it refers to better behaviour on the non-shadowed examples in dense areas. But what about low density? Why are these calculations not reflected in the results? 

- The conclusions are still very limited and do not lead to discussion. In turn, they bear little relation to the research carried out or possible links to the literature and research mentioned in the methodology.